# Guided visual search is associated with target boosting and distractor suppression in early visual cortex

Katharina Duecker [1,2] ✉, Kimron L. Shapiro[2], Simon Hanslmayr[3], Benjamin J. Griffiths[2,4], Yali Pan [2], Jeremy M. Wolfe[5,6] & Ole Jensen [2,7,8] ✉

Visual attention paradigms have revealed that neural excitability in higher-order visual areas is modulated according to a priority map guiding attention towards task-relevant locations. Neural activity in early visual regions, however, has been argued to be modulated based on bottom-up salience. Here, we combined Magnetoencephalography (MEG) and Rapid Invisible Frequency Tagging (RIFT) in a classic visual search paradigm to study feature-guidance in early human visual cortex. Our results demonstrate evidence for both target boosting and distractor suppression when the participants were informed about the task-relevant and -irrelevant colour (guided search) compared to when they were not (unguided search). These results conceptually replicated using both a magnitude-squared coherence approach and a General Linear Model based on a single-trial measure of the RIFT response. The present findings reveal that feature-guidance in visual search affects neuronal excitability as early as primary visual cortex, possibly contributing to a priority-map-based mechanism.

Visual search is a widely used paradigm, applied to operationalize the everyday task of finding a pre-defined stimulus (target) among distracting stimuli (distractors), for instance, a friend in a crowd. Search is more efficient for salient targets, and when low-level features of the target, e.g., colour or shape, are known to the observer, allowing for top-down guidance of attention. For example, when we know that our friend is wearing a yellow raincoat, we will pay less attention to people wearing blue jackets. The allocation of visual attention has long been suggested to involve a priority map: a representation of objects in the visual field in which object locations are weighted based on bottom-up saliency and top-down task-relevance[1–6]. Priority maps have become a central component of models of selective attention[1,7,8] and visual search[9,10]. In the example above, this map would assign high priority to objects containing the colour yellow and low priority to those containing the colour blue.

Evidence from behavioural, electrophysiological, and neuroimaging studies leaves little doubt that visual attention is guided by a mechanism akin to a priority map, whereby neural responses to the target are boosted, and responses to the distractors are reduced or suppressed[4,5,8,11–32]. Modulation of cortical excitability in accordance with a priority map has, for instance, been observed in electrophysiological

recordings in non-human primates from the frontal eye field and lateral intraparietal cortex[18,22,33] as well as V4[23]. Traditionally, it has been assumed that the primary visual cortex implements a bottom-up saliency map during visual search, while the top-down relevance of different locations for the task is encoded at later stages of the visual hierarchy[34–36]. However, attention to spatially separable stimuli has been shown to modulate neural activity in early visual areas; as e.g. quantified by the blood oxygenation level-dependent signal in functional magnetic resonance imaging (fMRI)[37–39], intracranial recordings in non-human primates[40] and event-related responses in electroencephalography (EEG) and magnetoencephalography (MEG)[41–45]. While these studies did not explicitly investigate visual search, they do suggest that attentional guidance benefits from recruiting V1; for instance, to utilize the high spatial resolution of the small receptive fields[24,46]. In this study, we use MEG in combination with Rapid Invisible Frequency Tagging (RIFT) in a classic visual search paradigm to test whether feature-guided search is associated with a modulation of neuronal excitability in early visual cortex. RIFT is a subliminal stimulation method to probe excitability of visual responses that has been shown to predominantly stimulate primary visual cortex[47–50], leaving endogenous oscillations unperturbed[47,51].

[1]Department of Neuroscience, Brown University, Providence, RI, USA. [2]Centre for Human Brain Health, School of Psychology, University of Birmingham, Birmingham, UK. [3]Centre for Cognitive Neuroimaging, School of Neuroscience and Psychology, University of Glasgow, Glasgow, UK. [4]School of Psychology, University of Nottingham, Nottingham, UK. [5]Brigham and Women's Hospital, Boston, MA, USA. [6]Harvard Medical School, Boston, MA, USA. [7]Department of Experimental Psychology, University of Oxford, Oxford, UK. [8]Oxford Centre for Human Brain Activity, Wellcome Centre for Integrative Neuroimaging, Department of Psychiatry, University of Oxford, Oxford, UK. ✉e-mail: katharina.duecker@gmail.com; ole.jensen@psych.ox.ac.uk

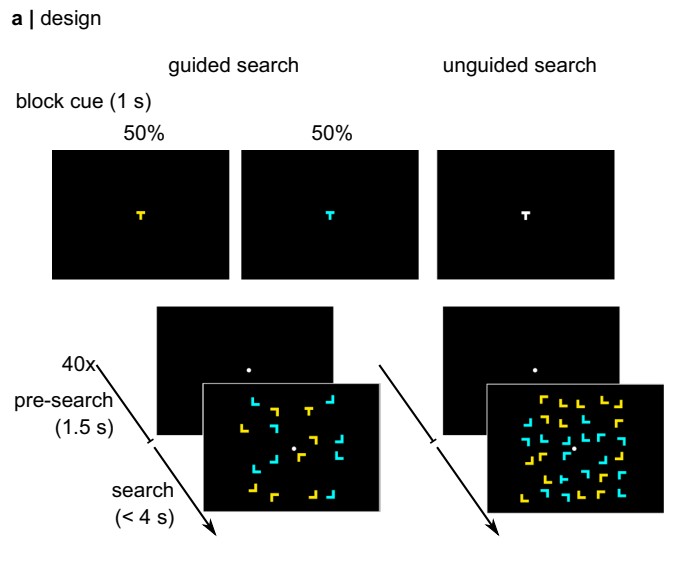

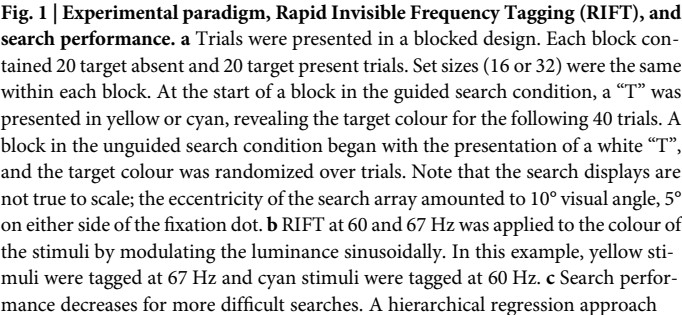

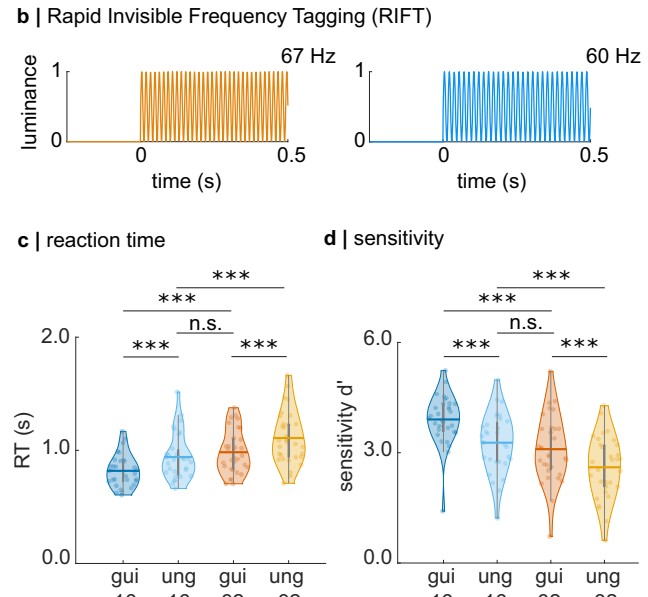

**Fig. 1 | Experimental paradigm, Rapid Invisible Frequency Tagging (RIFT), and search performance. a** Trials were presented in a blocked design. Each block contained 20 target absent and 20 target present trials. Set sizes (16 or 32) were the same within each block. At the start of a block in the guided search condition, a "T" was presented in yellow or cyan, revealing the target colour for the following 40 trials. A block in the unguided search condition began with the presentation of a white "T", and the target colour was randomized over trials. Note that the search displays are not true to scale; the eccentricity of the search array amounted to 10° visual angle, 5° on either side of the fixation dot. **b** RIFT at 60 and 67 Hz was applied to the colour of the stimuli by modulating the luminance sinusoidally. In this example, yellow stimuli were tagged at 67 Hz and cyan stimuli were tagged at 60 Hz. **c** Search performance decreases for more difficult searches. A hierarchical regression approach

reveals a significant main effect for set size ($\beta = 0.180$) and guided/unguided ($\beta = -0.138$). Indicating that larger set sizes are associated with slower responses, while guided searches are faster than unguided searches. Pairwise comparisons did not reveal any significant difference in reaction time between unguided search set size 16 and guided search set size 32 ($V = 134$, $z = 2.24$, $r = 0.4$, $p = 0.148$), suggesting that participants focused their search on task-relevant items in the guided search condition. **d** Analogously, for accuracy (as measured by $d'$), hierarchical regression reveals a significant main effect for set size ($\beta = -0.74$) and guided/unguided ($\beta = 0.56$), indicating that accuracy is higher in guided searches and for set size 16 compared to 32. Again, there is no significant difference in sensitivity for unguided search set size 16 and guided search set size 32 ($t(30) = 2.2$, $d = 0.2$, $p = 0.23$).

Behavioural studies of visual search often involve complex search displays with a large number of stimuli[52,53]. Electrophysiological approaches in humans and non-human primates, however, rely on spatially separable stimuli and therefore typically investigate visual search paradigms with smaller set sizes of up to six items (e.g. refs. 20,21,23,54–56). Other studies extrapolate the underlying mechanisms of visual search from experiments on selective attention, cueing the participant to attend to certain objects presented in a large field of stimuli[11,29,57]. Similarly, the neural dynamics of distractor suppression in humans and non-human primates are typically investigated in the context of actively ignoring a single, salient distractor[20,55,56,58–66]. In the context of these studies, it has long been debated whether the location of an expected singleton distractor can be suppressed in anticipation of the search display[67], with several studies arguing both for[39] and against[23] anticipatory distractor suppression in visual cortex.

To complement this body of work, we here leveraged MEG and RIFT to study feature guidance in a classic visual search paradigm with a relatively high number of 16 and 32 stimuli, in the tradition of early behavioural studies that motivated the hypotheses that search is guided by a map of the visual field[9,10,68] (Fig. 1a). The high spatial and temporal resolution of the MEG recording, paired with the high frequency range used for RIFT, allowed us to estimate both the source of the RIFT signal, and the latency of the attention effects. As we will show, the RIFT responses demonstrate that both target boosting and distractor suppression affect neuronal excitability as early as V1[69]. Based on the time course of the RIFT signal, we suggest that this modulation underlies downstream control from higher-order visual areas, such as the frontal eye field, lateral intraparietal cortex, and V4[18,22,23,33]. Considering the retinoptic organization of V1, our findings open the intriguing possibility that V1 may contribute to the implementation of a priority map to guide the search with high spatial resolution[46]. We offer suggestions on the mechanisms underlying this feedback control on V1 and how to study them.

## Results

Our experimental paradigm featured two search conditions (*guided* and *unguided search*) and two set sizes (16 and 32), presented in a block design (with a randomized order over all participants), with each block consisting of 40 trials (Fig. 1a). Participants were instructed to indicate if a single letter "T" was presented among several "L"s. In the *guided search* condition, participants were cued to the colour of the target "T" (either yellow or cyan) at the beginning of the block. Importantly, as only these two colours were used throughout the experiment, participants were able to infer the distractor colour from this cue. In the *unguided search* condition, a white "T" was presented at the beginning of the block, meaning the target and distractor colours were not cued, and the colour of the T was randomized over trials. Set size was kept constant within each block. The target and distractor colours were frequency-tagged by modulating their luminance sinusoidally, at 67 Hz and 60 Hz respectively (balanced over trials; Fig. 1b). Participants were instructed to perform the task while fixating on a centrally presented dot. Note that the 60 and 67 Hz flickers are invisible to the observer but modulate neuronal activity[70].

Based on the extensive literature on visual search, we predicted search performance to be worse (indicated by reaction time and accuracy) for more difficult searches, i.e. *set size 32* relative to *16* and *unguided* compared to *guided search*[9,10,71,72]. Indeed, these hypotheses were confirmed by a hierarchical regression approach applied to the average reaction time and sensitivity ($d'$) for each participant. Reaction time was significantly increased for higher set size ($\beta = 0.180$) and was reduced for *guided* compared to *unguided search* ($\beta = -0.138$, see Supplementary Analyses). A Wilcoxon signed-rank test revealed no significant difference between *unguided search, set size 16* and *guided search, set size 32*, indicating that the difficulty of these searches was similar as would be expected if colour guidance could render half of the distractors irrelevant in the *guided search, set size 32* condition

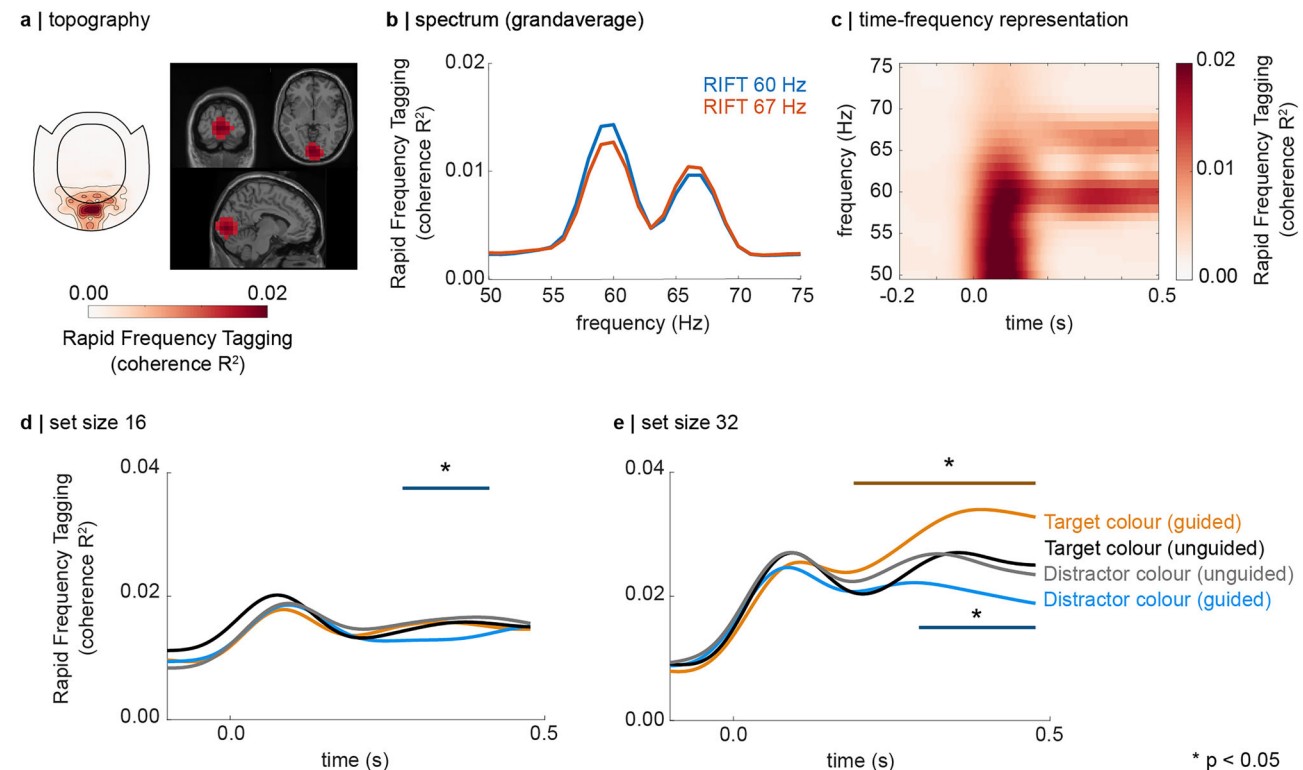

**Fig. 2 | Rapid Invisible Frequency Tagging (RIFT) responses reflect a target boosting and distractor suppression. a** (left) Topographic representation of the 60 Hz RIFT signal (uncombined planar gradiometers), averaged over participants in the 0.1–0.5 s interval ($t = 0$ s is the onset of the search display). The RIFT response is confined to the occipital sensors. (right) Source modelling demonstrates that the RIFT response was primarily generated in the early visual cortex. The source grid has been masked to show the 1% most strongly activated grid points (MNI coordinates [0 −92 −4]). **b** Grand average spectrum, obtained by averaging over the participant-specific sensors of interest, indicating peaks at the 60 and 67 Hz stimulation frequency. **c** Grand average of the time-frequency representation of coherence between the MEG sensors and the RIFT signal, demonstrating an early, unspecific response in the gamma-band, followed by narrow-band responses at the stimulation frequency. **d** Set size 16. The responses to the distractor colour are significantly reduced for guided compared to unguided search ($p < 0.05$; multiple comparison controlled using a cluster-based permutation test in the 0.1–0.5 s interval). There is no evidence for target boosting for this set size. **e** Set size 32. The RIFT responses to the guided target colour are significantly enhanced and the responses to the guided distractor colour are significantly reduced compared to the unguided search condition ($p < 0.05$; cluster-based permutation test).

($V = 134$, $z = 2.24$, $r = 0.4$, $p = 0.148$, see Supplementary Tables 2 and 4 and Fig. 1c). Similarly, sensitivity decreased for the higher set size ($\beta = -0.74$) and increased for guided vs. unguided search ($\beta = 0.56$), with no difference between *unguided search, set size 16* and *guided search, set size 32* as indicated by an independent sample t test ($t(30) = 2.2$, $d = 0.2$, $p = 0.23$). These behavioural findings demonstrate that the participants used the colour cue at the beginning of the block to focus their search on the target colour.

## RIFT responses indicate target boosting and distractor suppression

RIFT elicited brain responses at the respective stimulation frequencies that were detected in a small number of MEG sensors over the occipital cortex (Fig. 2a, left, see Supplementary Fig. 1 for individual topographic representations per participant). Source modelling based on dynamic imaging of coherent sources (DICS) demonstrated that the responses emerged from early visual regions (V1, MNI coordinates [0 −92 −4], Fig. 2b). The spectrum in Fig. 2b indicates that the coherence between the MEG sensors of interest and the RIFT signal is frequency specific, i.e. the response to the 60 Hz RIFT signal is strongest at 60 Hz and vice versa for the 67 Hz signal. The grand average time-frequency representation of coherence in Fig. 2c indicates an evoked gamma-band response at the onset of the search display, followed by narrow-band responses at the RIFT frequencies that are sustained until the end of the search.

As outlined in our pre-registration (https://osf.io/vcshj), we hypothesized that the RIFT response reflects a priority-map-based search strategy, indicating target boosting and distractor suppression in the *guided search* condition. Figure 2d, e show the RIFT response quantified by the coherence ($R^2$) between the MEG response (*RIFT sensors of interest*) and the frequency tagging signal, averaged over participants (see "Methods" for details on the RIFT analysis). Comparison to Fig. 2c demonstrates that the immediate increase in coherence after the onset of the search display reflects a broadband evoked response, rather than the frequency-specific flicker signal. The RIFT responses for *set size 16* were noticeably weaker than in the set size 32 condition, but indicated significantly reduced responses to the distractor colour for *guided* compared to *unguided search*, which we interpret as evidence for distractor suppression (Fig. 2d, compare blue line to the average of the grey and black lines, $p < 0.05$; multiple comparisons were controlled using a Monto-Carlo cluster-based dependent sample t test on the 0.1 to 0.5 interval, 1000 permutations).

For *set size 32*, we find that the RIFT responses to the target colour in the *guided search* condition were significantly enhanced compared to the *unguided search* condition (compare orange line to average of the black and grey lines in Fig. 2e, $p < 0.05$; cluster-based test as described above). This suggests a boosting of the neuronal excitability to all items sharing the known target colour. Importantly, the responses to the distractor colour when comparing *guided* to *unguided search* were again significantly reduced, providing evidence for distractor suppression (Fig. 2c, $p < 0.05$; 1000 permutations). Our findings demonstrate that knowledge about the target and distractor colour in the *guided search* condition results in a modulation of the RIFT response consistent with the concept of a priority map, whereby target representations are boosted, and distractor representations are suppressed. Furthermore, these results demonstrate that

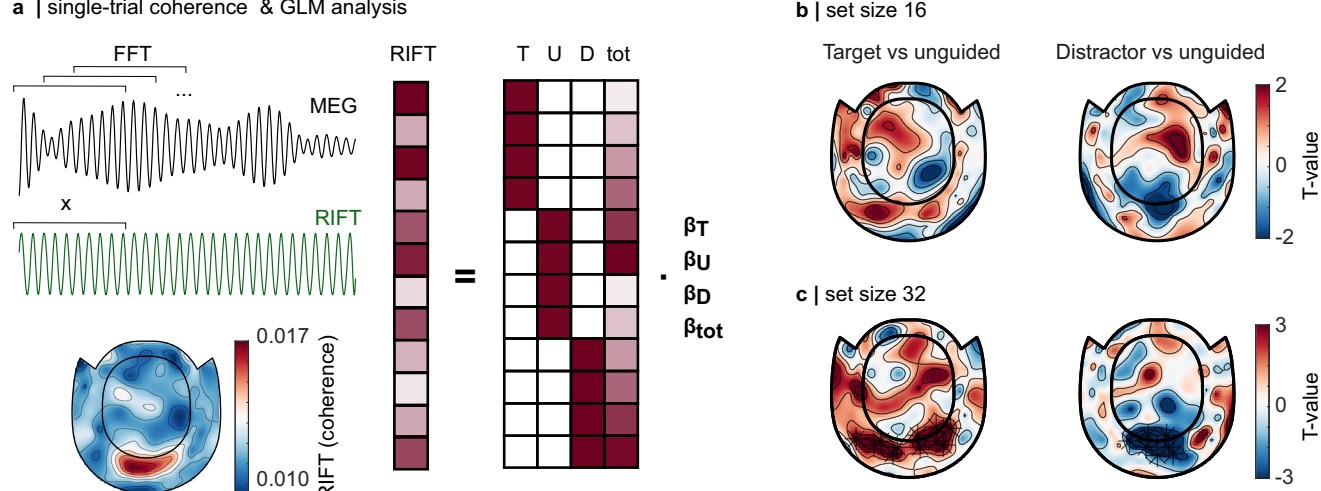

**a |** single-trial coherence & GLM analysis

**b |** set size 16

**c |** set size 32

**Fig. 3 | Quantifying the RIFT response at the single-trial level. a** (left, top) The coherence between the signal-trial MEG data and RIFT signal was quantified using a sliding-window FFT approach, whereby the coherence was estimated by averaging over the 0.1 s in the 0.2–0.5 s interval (moved in steps of 0.025 s). (left, bottom) The topography of the coherence (combined planar gradiometers) suggests a response in the occipital sensors. (right) The RIFT response to the target and distractor was then concatenated into one vector, and submitted to a GLM with the factors target colour (T), unguided (U), distractor colour (D), and time-on-task (tot). **b** The contrast between the regressors associated with the target colour and unguided, and between distractor colour and unguided, was compared to 0 using a cluster-based permutation test (5000 permutations). The model fitted to the set size 16 conditions yielded no significant results but suggested reduced responses for distractors compared to unguided stimuli ($p = 0.08$). **c** The model fitted to the set size 32 conditions replicated the magnitude-squared coherence results reported above, with a significantly stronger response to the target colour compared to unguided and a significantly reduced response to the distractor colour.

RIFT is suitable to measure the neuronal excitability associated with the priority map in visual search.

## Investigating target boosting and distractor suppression at the single-trial level using a Generalized Linear Model (GLM)

The magnitude-squared coherence results described above quantify the degree to which the variance in the MEG signal is accounted for by the RIFT signal[73,74]. Coherence is therefore a more interpretable measure for RIFT than spectral power, which is likely to be confounded by changes in broadband and/or oscillatory activity in the gamma-band. One caveat of magnitude-squared coherence, however, is that it requires averaging over observations[75]. A single-trial quantification of the RIFT response is desirable, as it opens opportunities for several different analytic approaches, for instance, a GLM approach, used to link changes in oscillatory activity to experimental manipulations and behaviour[76,77]. We therefore sought to confirm the reported results using a single-trial quantification of the RIFT response. Furthermore, using this approach, we were able to account for effects of task duration, and thus neural adaptation to the repeated presentation of the search display[78].

First, we filtered each trial in the 30–80 Hz band using a two-pass Butterworth filter. The single-trial RIFT response was quantified using a windowing approach, whereby a sliding window of 0.1 s, multiplied with a Hanning taper, was moved in steps of 0.025 s over the 0.2–0.5 s interval following the onset of the search display in each trial (Fig. 3a, top). This interval was chosen to avoid confounds with the broadband gamma response to the onset of the display Fig. 2c. For each window, coherence between the MEG and RIFT signal was estimated based on the Fast Fourier Transform (FFT; zero-padded to 512 samples), and averaged over all windows to obtain one coherence value per trial (as implemented by the *mscohere* function in MATLAB). The resulting grand average topography is shown for the combined planar gradiometers in Fig. 3a (bottom), showing the strongest coherence in a small set of occipital sensors.

To investigate target boosting and distractor suppression while accounting for task duration, we concatenated the coherence values for the target and distractor colours into one long vector. The effect of task duration and stimulus was then investigated using a GLM approach, guided target, guided distractor, and unguided stimuli modelled as separate regressors $\beta_T$, $\beta_D$, and $\beta_U$, respectively (Fig. 3a, right). Additionally, time-on-task (tot) was integrated into the design matrix based on the trial index for the concatenated RIFT responses for the targets and the distractors, and ranged from 0 to 1. To account for the inter-individual variability in the RIFT signal[47], we fit the model to each participant individually and calculated the $T$-values for target boosting and distractor suppression based on the contrast between $\beta_T$ and $\beta_U$ and between $\beta_D$ and $\beta_U$ (see "Methods"). The resulting $T$-values in each MEG sensor, and for each participant, were then compared to 0 using a dependent-sample cluster-based permutation $t$ test.

While a model including all set size 16 trials did not yield a significant effect of stimulus type, we found that the reduced response to the distractors (indicated by a negative $T$-value for the contrast between $\beta_D$ and $\beta_U$ over the occipital sensors) showed a trend effect with $p = 0.08$ for the cluster. These less robust effects are likely explained by a reduced signal-to-noise ratio for the set size 16 condition, and a relatively small effect of distractor suppression for set size 16, as indicated by Fig. 2d (Fig. 3b, right). There was no indication of a significant difference between $\beta_T$ and $\beta_U$ (Fig. 3b, left). When fitting the GLM only to trials with *set size 32*, we found that the $T$-values associated with the $\beta_T$ and $\beta_U$ contrast were significantly larger than 0, indicating that the RIFT responses to the target were significantly enhanced compared to the *unguided search* condition ($p < 0.05$, Fig. 3c, left). Likewise, the contrast between $\beta_D$ and $\beta_U$ showed a significantly weaker response to the distractors compared to unguided search ($p > 0.05$, Fig. 3c, right). These findings are generally in line with the coherence results presented above.

Overall, the GLM approach replicated the results of the magnitude-squared coherence. As such, the presented method can be used as a complementary measure to link RIFT responses to behaviour or other neural measures of interest, without the need to collapse over trials.

## Relevance of observed RIFT modulation for behaviour not established

Using two complementary approaches, we demonstrate that the RIFT response is modulated in line with a priority-map-based mechanism. Next, we sought to test whether this modulation is relevant for performance. For the magnitude-squared coherence that requires averaging over trials, we sorted the trials in each condition based on a median split on reaction time to

compare the coherence between fast and slow trials (see Supplementary Analyses and Supplementary Fig. 2a, b). For the GLM approach, we selected the trials in the *guided search* condition (both set sizes 16 and 32) and fitted separate GLMs to the RIFT responses associated with the target and distractor colour, including the regressors constant, tot, and reaction time. We hypothesized that fast trials might be associated with stronger target boosting and/or distractor suppression, but neither the magnitude-squared coherence nor the GLM approach indicated this to be the case (see Supplementary Fig. 2).

### RIFT modulation is not explained by eye movements

Even though participants were instructed to perform the search task without making ballistic eye movements, an argument could be advanced that the observed modulation of the RIFT responses results from an eye movement bias towards the target. Excluding all trials in which saccades and microsaccades occur was not feasible, as the participants appeared to be unable to keep their eyes still for the duration of the 1.5 s baseline and search period of a minimum of 0.5 s. This is likely to be the case as fixational eye movements are needed to avoid visual fading caused by neural adaptation to a stabilized retinal image[79,80]. We therefore sought to ensure that these saccades did not reflect a gaze bias towards the target that could explain the reported effects on target boosting and distractor suppression.

To this end, we first compared the number of saccades and eye blinks parsed online by the EyeLink® eye tracker (see "Methods") based on a median split on reaction time (see Supplementary Analyses). As demonstrated in Supplementary Fig. 3, fast trials were not associated with a significantly higher or lower number of blinks or saccades compared to slow trials. While we did ensure that the search stimuli of each colour were evenly randomized over the display, we further tested whether the participants tended to move their gaze towards the target colour. This gaze bias was identified by binning the single-trial eye-tracking data into 0.1 s intervals and counting how often the gaze was closest to a stimulus in the target colour. The number of occurrences when the gaze was closest to the target colour was then divided by the total number of bins in the trial and averaged over all trials. As shown in Supplementary Fig. 3c, the gaze bias appeared to average at about 0.5 (i.e. 50% of the time bins within a trial), indicating that the gaze was in the vicinity of target and distractor stimuli for about equal amounts of time. Importantly, there was no difference in gaze bias for fast vs. slow trials, suggesting that participants generally followed the instructions and solved the task without moving their eyes.

Finally, the heatmap presented in Supplementary Fig. 3d demonstrates that participants largely followed the instructions and kept their gaze within one degree visual angle of the fixation cross (the calibration threshold of the eye tracker, indicated by the inner box). In line with previous research, microsaccades occurred predominantly along the horizontal plane[79].

We conclude that the target boosting and distractor suppression observed in the RIFT response reflects a modulation of the excitability of early visual neurons, which underlies feature-based attention and not eye movement.

## Discussion

We used RIFT in combination with MEG as a novel approach to probe neuronal excitability in the visual cortex, to investigate feature-guided visual search. In the *guided search* condition, the target colour was cued, whereas in the *unguided search* condition, the target colour was unknown. As expected, search performance was reduced for higher set sizes and for *unguided* compared to *guided* search; the latter confirming that participants used the colour cue at the beginning of the block to guide their search. Importantly, the RIFT responses revealed in the *guided search* condition, *set size 32*, demonstrated an increase in neuronal excitability in early visual cortex associated with the target colour and a suppression associated with the distractor colour. As we will argue below, these results suggest that the early visual cortex may play an important role in a priority-map-based account for the purpose of guiding complex visual search based on known target and distractor features. This mechanism is likely to underlie top-down control.

Our work complements previous electrophysiological recordings in humans and non-human primates investigating visual search paradigms with smaller set sizes of up to six, spatially distinguishable, items[20,21,23,54–56]. Feature-based attention, on the other hand, has been studied using visual flickers applied to moving stimuli[11,29,57]. In contrast to these works, we employed a complex visual search display with a large set size, as traditionally used in psychophysical research[9,10,68]. This allows a more direct test of the theory that visual search is guided by feature-based attention[9,10].

Using MEG inverse modelling, we localized the source of the RIFT response to the early visual cortex. This is consistent with recent studies suggesting that rhythmic responses to a high-frequency flicker do not propagate meaningfully beyond V1/V2 (ref. 47; also see refs. 49,81). We theorize that this priority-map-based account involves the recruitment of retinotopically organized V1 neurons, effectively utilizing their small receptive field size to guide visual search with high spatial precision[82,83].

The modulation of the RIFT signal can be observed at about 200 ms after search display onset. This time course is congruent with the observation that guidance by colour takes about 200–300 ms to be effective[84], and further conforms with the latency of previously observed effects of attention on neural activity in V1[85]. Electrophysiological recordings in non-human primates have shown target boosting and distractor suppression in the frontal eye field and lateral intraparietal cortex about 90 ms after stimulus onset[18,22,33] and after about 110 ms in V4[23]. In light of these findings, we propose that neuronal excitability in V1 may be modulated by higher-order visual areas through feedback connections, as shown in spatial attention paradigms[46,86–90]. Our findings are intriguing in that they suggest that the excitability of colour-responsive, retinotopically organized neurons in early visual cortex can be modulated by visual areas with a lower spatial resolution. While the importance of feedback connections to V1 in visual search and attention has been discussed in the context of the Reverse Hierarchy Hypothesis[46], feature-guidance in early visual regions has been argued to underlie saliency of the stimulus, or a serial search of the display[34,46]. We propose that the presented findings suggest that feedback connections modulate the excitability of the colour-sensitive early visual neurons in parallel. This hypothesis could be tested based on a modified version of the experiment with spatially separable stimuli in the target and distractor colour and the guided/unguided feature cue. Neuropixels recordings in non-human primates would allow simultaneous recordings in multiple cortical areas and could be used to investigate the spike rates of neurons responding to stimuli sharing the target and distractor colour. Based on our findings, we predict that the spike rates of neurons with different receptive fields in V1 would be modulated in parallel to boost potential target locations and suppress distractors.

To conclude, we argue that the timing and spatial specificity of our results are most consistent with the idea that higher-order areas recruit early visual neurons to benefit from their high spatial resolution during complex visual search. This suggests that the early visual cortex partakes in the execution of a priority-map-based mechanism, which has been argued to emerge through a collaboration between several sensory and cognitive processes[9].

### Limitations and outlook

Target boosting and distractor suppression have been argued to be implemented by two distinct mechanisms[54,58,91]. In this study, participants were able to infer the distractor colour from the cue provided at the beginning of the block, thus preventing us from disentangling these mechanisms. In future studies, it would be useful to use RIFT in a paradigm relying solely on distractor inhibition[92], in which the participants are only informed about the distractor colour, while the target colour varies. This would clarify how distractor suppression is implemented when the target colour is unknown.

It is strongly debated whether known distractors can be suppressed in anticipation of the search display[67,93]. Anticipatory distractor suppression is traditionally studied using spatially distinct, salient distracting stimuli[23,59]. We here aimed to study the modulation of neural excitability associated with feature-guided and unguided search amongst a high number of stimuli, and

therefore refrained from including singleton distractors. The time course of the RIFT response presented here does however not indicate any evidence for anticipated suppression of the known distractor colour. However, the gamma-band response to the onset of the search display (see Fig. 2) complicates conclusions about the onset of the distractor suppression and target boosting in these data. As such, the current experiment was not optimized to answer the question of whether known distractors can be suppressed in anticipation. However, as noted above, the observed evidence for distractor suppression for both set size 16 and 32 suggests that ignoring irrelevant stimuli may be an important strategy for efficient visual search.

While the modulation of the RIFT response is clearly observed for the set size 32 condition, the results in the set size 16 condition were less robust. This could be due to a lower number of pixels flickering, resulting in a reduced signal-to-noise ratio. Alternatively, the set size 16 condition might not benefit from guidance in early visual cortex, as the larger distance between the stimuli might not necessitate a retinotopic resolution (see above). Moreover, the large distance between the stimuli might reduce the need for target boosting and distractor suppression in this condition. Previous EEG work has linked the amplitude of the N2Pc (indicative of enhancement) and Pd (associated with distractor suppression) component of the ERP to the proximity of targets and distractors, suggesting that a small distance between task-relevant and irrelevant stimuli increases the need for target boosting and distractor suppression. Indeed, a recent MEG study has demonstrated that enhanced responses to a target stimulus at a known location are associated with reduced responses to a nearby distractor[69]. Variations of the current set size 16 condition with varying distances between the search stimuli may serve to disambiguate the question of whether the observed results are due to insufficient signal-to-noise ratio or the difficulty of the search.

## Conclusion

In conclusion, our work demonstrates that guided search is associated with a modulation of neuronal excitability in early visual regions according to a priority map. As we have argued above, the retinotopic organization of early visual regions and the onset of the effect after about 200 ms, suggest that this modulation underlies top-down control from higher-order visual regions. While the presented results do not allow conclusions about the source of the priority map, they suggest that feature-guidance in visual search plays a role in its implementation.

## Methods

### Experimental design and stimuli

**Task**. We applied RIFT in a classic visual search paradigm to probe the neuronal excitability to the target and distractor colour in *guided* and *unguided search*. The participants' task was to indicate whether a cyan or yellow letter "T" was present or absent among several cyan and yellow "Ls" (Fig. 1a). Each participant completed 24 blocks of 40 trials each. At the beginning of a block in the *guided search* condition, a yellow or cyan letter "T" was presented in the centre of the screen, indicating the colour of the target for the following block (Fig. 1a). In blocks in the *unguided search* condition a white "T" was shown before the trial, and the colour of the target in the search display, if present, was randomly chosen to be cyan or yellow over trials. The set size of each 16 or 32 items was kept constant within each block. As such, each participant completed 240 trials of *guided* and *unguided* search, respectively, for each set size (960 trials total). Every trial started with a 1.5-s baseline interval in which a white fixation dot was presented in the centre of the screen. The trials were terminated with the participants' button press, or automatically after 4 s. The button press was followed by a black screen, presented for 500 ms, before the start of the pre-search interval of the following trial. All participants completed four practice blocks consisting of 10 trials each before the experiment. Participants were instructed to find the target without moving their eyes. The experiment and MEG recording were paused every 10 min, and participants were encouraged to rest their eyes and move their heads.

**Display physics**. The stimuli were presented using a Propixx lite projector (VPixx Technologies Inc., Quebec, Canada), set to a refresh rate of 480 Hz. The luminance of the yellow and cyan stimuli in the search display was modulated sinusoidally, respectively at 60 and 67 Hz (Fig. 1b, target and distractor colours, tagging frequencies, and set sizes were randomized within participants). The stimuli consisted of horizontal and vertical bars with a width and height of 1° visual angle, arranged in a search grid of 10° × 10°, i.e. 5° in each direction from the fixation point. The search display was created using the Psychophysics Toolbox version 3[94] in MATLAB 2017a (The Mathworks, Natick, MA, USA).

### Apparatus for data acquisition

The MEG data were acquired using a MEGIN Triux (MEGIN Oy, Espoo, Finland), with 204 planar gradiometers and 102 magnetometers at 102 sensor positions, housed in a magnetically shielded room (Vacuumschmelze GmbH & Co, Hanau, Germany). Data were filtered online between 0.1 and 330 Hz using anti-aliasing filters and then sampled at 1000 Hz. The dewar orientation was set to 60° to allow the participants to comfortably rest their heads against the back of the sensor helmet, optimizing the recording of the neuromagnetic signals in the occipital cortex.

The three fiducial landmarks (nasion and left and right periauricular points), the participant's head shape (>200 samples), and the location of four head-position-indicator (HPI) coils were digitized using a Polhemus Fastrack (Polhemus Inc., Vermont, USA) prior to the recording. The location of the HPI coils was acquired at the beginning of each new recording block, but not continuously throughout the experiment.

The RIFT signals at 60 and 67 Hz were further applied to two squares at the outer corners of the screen and recorded using two custom-made photodiodes (Aalto NeuroImaging Centre, Aalto University, Finland), connected to the MEG system.

Eye movements and blinks were tracked using an EyeLink® eye tracker at a sampling rate of 1000 Hz (SR Research Ltd, Ottawa, Canada), positioned at the minimum possible distance from the participant. The conversion of the EyeLink® Edf files was done with the Edf2Mat Matlab Toolbox designed and developed by Adrian Etter and Marc Biedermann at the University of Zurich. For the online saccade detection, we used conservative thresholds that allow detection of eye movement as small as 0.3 degrees: 0.1 degrees for the motion, 22 degrees/s for velocity, and 3800 degrees/s² acceleration (see EyeLink® manual). The identified saccades were used for the Supplementary Analyses of ocular artefacts and gaze bias described below.

The T1-weighted anatomical scans were obtained using a whole-body 3-Tesla Philips Achieva scanner (echo time TE = 0.002 s, repetition time TR = 2 s).

### Participants

This study was carried out in accordance with the Declaration of Helsinki and the COVID-19-related safety measures at the University of Birmingham in place between April 2021 and January 2022. All ethical regulations relevant to human research participants were followed, and ethical approval by the University of Birmingham ethics committee was obtained prior to any data collection associated with this study. A telephone screening was conducted 48 h before the experiment to ensure that all participants were safe for MRI and free of COVID-19 symptoms. Forty-eight volunteers with no history of neurological disorders gave written informed consent prior to participating in the MEG experiment or structural MRI scan. The participants' colour vision was assessed prior to the experiment using 14 Ishihara plates[95]. Participants for whom the eye tracking recording was missing due to technical errors were not considered for the analysis ($N = 6$). Three additional participants were excluded as their button presses often extended into the following trials (in 160–300 trials), resulting in a total sample size of $N = 39$. Participants who did not show a significant tagging response ($N = 8$) were excluded at a later stage (see RIFT response sensor selection below), leaving 31 data sets (20 females, see below).

## Behavioural performance

The participants' performance on correctly detecting the presence and absence of the target was quantified based on average reaction time and perceptual sensitivity ($d'$), calculated as:

$$d' = z(H) - z(FA) \tag{1}$$

with $z(H)$ being the $z$-scored portion of hits in target present trials and $z(FA)$ being the $z$-scored portion of false alarms in target absent trials.

## MEG pre-processing

Signal Space Separation (SSS, "Maxfilter") implemented in MNE Python was applied to suppress magnetic signals emerging from sources outside the participant's brain. The remaining pre-processing of the MEG data, frequency and source analyses, and cluster-based permutation test were performed using the Fieldtrip toolbox[96] in MATLAB 2019b. Statistical analyses of the behavioural and eye tracking data were carried out in RStudio 1.1.456 with R version 3.6.1. (The R Foundation for Statistical Computing).

Faulty sensors were identified and corrected prior to the SSS using MNE python. The filtered data were divided into intervals of 4.5 s, starting 2.5 s before, and extending to 2 s after the onset of the search display in each trial. Semi-automatic artefact rejection was performed on the 4.5 s intervals, by manually identifying and rejecting epochs with a comparably high variance, separately for gradiometers and magnetometers. Independent component analysis was used to suppress oculomotor and cardiac artefacts based on the 68 components that were identified for each participant. Trials with unreasonably short reaction times of up to 200 ms, as well as trials without a response were rejected[97].

## RIFT response sensor selection

The MEG sensors containing a reliable frequency tagging response were identified using nonparametric (Monte Carlo) statistical testing, proposed by Maris and Oostenveld[98]) and implemented in the Fieldtrip toolbox. The pre-processed data were divided into a baseline (0.7–0.2 s before stimulus onset) and stimulation interval (0.5 s following the onset of the search display). Coherence between a given MEG sensor and the 60 Hz photodiode signal over trials was estimated separately for the pre-search and the search interval. The difference between the coherence in the baseline and search interval was $z$-transformed using the following equation:

$$Z = \frac{(\tanh^{-1}(|\text{coh}_{\text{search}}|) - \text{bias}) - (\tanh^{-1}(|\text{coh}_{\text{bsl}}|) - \text{bias})}{\sqrt{2 * \text{bias}}} \tag{2}$$

whereby $\text{coh}_{\text{search}}$ and $\text{coh}_{\text{bsl}}$ are the coherence between the respective MEG sensor and the photodiode at 60 Hz during the search and pre-search interval, respectively. The bias is calculated as $\text{bias} = \frac{1}{2n-2}$ with $n$ being the number of trials.

The statistical significance of the $z$-transformed coherence difference (the empirical $z$-value) was estimated using a permutation procedure. To this end, a null distribution for the empirical $z$-value was estimated by generating 10,000 random permutations of the trial labels and calculating the $z$-values for the shuffled pre-search and search interval, again using Eq. (2). If the coherence difference obtained for the unshuffled data in the respective sensor was larger than 99% of the null distribution, the sensor was considered to show a significant tagging response at a 1% significance level. This procedure was completed for a total of 81 occipital and occipito-parietal sensors to identify the sensors of interest for each participant. Thirty-one out of 39 participants had at least one significant gradiometer. As only 27 participants showed a significant response in at least one magnetometer, only gradiometers were considered for the sensor and source analyses. In total, we used the data from 31 volunteers for further analyses (20 females; aged 23.4 years ± 3.18). All participants were right-handed according to the Edinburgh Inventory (augmented handedness score: M = 84.08; STD = 14.37 (ref. 99).

## RIFT response magnitude

**Magnitude-squared coherence.** For the offline analyses, we replaced the photodiode signals with a perfect sine wave with the same phase as the RIFT signal, extending into the baseline interval. The magnitude of the RIFT response was quantified by calculating the spectral coherence between the MEG sensors of interest, identified as described above, and RIFT signal. The data were bandpass-filtered using a two-pass windowed-sinc finite-impulse response filter at 60 and 67 ± 3.5 Hz, respectively. The analytic signal was obtained from the filtered data using the Hilbert transform. The spectral coherence was then calculated as[73]:

$$\text{coh}_{\text{meg,diode}}(t) = \frac{|n^{-1} \sum_{k=1}^{n} m_{\text{meg}}(t) m_{\text{diode}}(t) e^{i\phi(t)}|}{(n^{-1} \sum_{k=1}^{n} |m_{\text{meg}}|)(n^{-1} \sum_{k=1}^{n} |m_{\text{diode}}|)} \tag{3}$$

with $m_{\text{meg}}$ and $m_{\text{diode}}$ being the analytic MEG and RIFT amplitude, respectively, $\varphi$ being the phase difference between the two signals, and n being the number of trials. To obtain the coherence to the RIFT signal of the target colour, for instance, we split the data into trials in which the target colour was tagged at 60 and 67 Hz, and calculated the coherence separately over these trials. Afterwards, the coherence was averaged over the two frequencies. Note that for the spectra and time-frequency representation presented in Fig. 2b, c, showing the coherence between the MEG and RIFT signal at frequencies from 50 to 75 Hz we added a small amount of noise (with an amplitude of 0.05) to the perfect sine wave, to avoid division by 0 (which is the power spectral density of the perfect sine wave at frequencies different from 60 and 67 Hz).

**Generalized Linear Model.** As the mean-squared coherence averages over observations (see Eq. (2)), we next sought to investigate the RIFT response at the single-trial level. While single-trial measures have the disadvantage of a reduced signal-to-noise ratio and loss of temporal information, they allow correlations with behavioural measures and confounding variables that change over time. Here, we used the single-trial measure of coherence in conjunction with a GLM to account for effects of tot on the RIFT signal.

The single-trial MEG data were first filtered with a two-pass Butterworth filter at 30–80 Hz, in the −2 to 4 s interval before and after the onset of the search display. This generous length was chosen for the epochs to avoid edge effects of the filter in the time window of interest (during the search). The coherence between the MEG signal in each trial and the photodiode was then estimated in the 0.2–0.5 s interval after the onset of the search display using Welch's method, with MATLAB's built-in function *mscohere*. This method divides the signal in each trial into segments (here 0.1 s with a 75% overlap), multiplies each segment with a window function (here Hanning taper), applies the FFT to each window, and calculates the coherence based on the cross-spectral and power-spectral densities of the MEG and photodiode signal.

The resulting single-trial RIFT measures for the target and distractor colour were then concatenated for each participant and further investigated using a GLM approach (for an in-depth description of the methods, see Quinn et al.[77]).

The single-trial correlation in each gradiometer ($\text{RIFT}_{\text{grad}}$) was modelled as a linear function of stimulus identity according to

$$\text{RIFT}_{\text{grad}} = XB_{\text{grad}} + e$$

$\text{RIFT}_{\text{grad}}$ contained the RIFT responses in the gradiometers, concatenated for targets and distractors, respectively. $X$ is the design matrix with four columns for the guided target, guided distractor, and unguided stimuli, respectively, as well as time-on-task (tot). The target column contained values of 1 for each row with a RIFT response to a guided target, and 0 for the unguided stimuli and distractors. Column 2 and 3 followed the same logic and were set to 1 for the unguided stimuli and distractors, respectively, and 0 everywhere else. The values of the tot regressor ranged from 0 to 1.

$B_{grad}$ contains the regressors associated with the RIFT response to the target colour, unguided stimuli, and distractors gradiometers for each gradiometer, as well as time-on-task, and was estimated by multiplying the Moore–Penrose pseudo-inverse of $X$ with $RIFT_{grad}$[77].

$$\widehat{B} = X^{+}RIFT$$

Note that we dropped the grad index for readability.

Following conventional approaches in fMRI data analysis[100], we calculated the Contrast of Parameter Estimates (cope) for the GLM are calculated by multiplying the estimated regressors with the contrast vector $C$

$$cope = C\widehat{B}$$

whereby $C_{target} = \begin{bmatrix} 1 \\ -1 \\ 0 \\ 0 \end{bmatrix}$ was multiplied with $\hat{B}$ to quantify target boosting

and $C_{distractor} = \begin{bmatrix} 0 \\ -1 \\ 1 \\ 0 \end{bmatrix}$ was used to quantify distractor suppression. The

variance of the respective contrast of the model was calculated as

$$varcope = diag(C(X^TX)^{-1}C)\sigma^2$$

with $\sigma^2$ being the variance of the residuals. The $t$ values for each contrast were then calculated as

$$t = \frac{cope}{\sqrt{varcope}}$$

We used the $t$ values rather than the estimated regressors, as they account for the variance of the contrasts[77]. The resulting $t$ values for each sensor were then compared to 0 using a one-tailed cluster-based permutation $t$ test with 5000 permutations.

### Source localization

The anatomical sources of the RFT response were estimated using the DICS[101] beamformer, implemented in the Fieldtrip toolbox[96].

**MEG lead field.** To calculate the MEG lead field, we first aligned the fiducial landmarks in the individual T1-weighted images with the digitized points taken prior to the experiment. The coordinate system of the participant's T1-weighted scan was then automatically aligned to the digitized head shape using the iterative closest point algorithm[102], implemented in the Fieldtrip toolbox, and corrected manually as necessary. For the two participants for whom there was no T1 scan available, the digitized fiducial landmark and head shape were aligned with a standardized template brain provided with the Fieldtrip toolbox.

Next, the brain volume was discretized into a source grid of the equivalent current dipoles by warping each participant's realigned anatomical scan to the Montreal Neurologic Institute (MNI) coordinate system, using a template MRI scan, and an equally spaced 8 mm grid, with 5798 locations inside the brain. The lead field was then estimated at each point in the source grid using a semi-realistic headmodel[103].

**Dynamic imaging of coherent sources.** The spatial filters of the DICS beamformer were calculated as a function of the forward model (estimated using the lead field matrix) and the cross-spectral density matrix of the sensor data. Here, we used the cross-spectral matrix of the gradiometers only. The SSS ("Maxfilter") caused the data to be rank deficient, making the estimate of the sensor cross-spectral density matrix unreliable. To ensure numerical stability, we calculated the truncated singular value decomposition (SVD) pseudoinverse[104,105] of the sensor cross-

spectral density matrix. This method decomposes the covariance matrix using SVD, selects a subset of singular values (the subset size is defined by the numerical rank) and calculates a normalized cross-spectral density matrix using this subset. The spatial filters are then estimated based on the normalized cross-spectral density matrix using unit-noise gain minimum variance beamforming[105,106].

To estimate the cross-spectral density matrix for the RIFT response, we first extracted data segments from 0 to 0.5 s (the minimum reaction time for all participants). The complex cross-spectral density between the signal in the (uncombined) planar gradiometers and the RIFT signal was computed based on the Fourier-transformed data segments (Hanning taper, separately for the 60 and the 67 Hz photodiode signal). The cross-spectral density matrices were used to estimate the forward model to create a spatial filter for each frequency. The spatial filters were then applied to the cross-spectral density matrix to estimate the RIFT response as the coherence between each point in the source grid and the photodiode signal.

**Statistics and reproducibility.** The effect of search condition on behaviour shown in Fig. 1 was tested for statistical significance using a linear mixed model with a hierarchical regression approach, whereby the full model was compared to a model containing only a subset of the regressors of interest (see Supplementary Material). This approach revealed significant main effects for both set size and condition. Pairwise comparisons using the Wilcoxon signed-rank test (for reaction time) and dependent sample $t$ test (for sensitivity) were corrected for multiple comparisons using the Hedges and Bonferroni corrections, respectively.

The effect of search condition and stimulus type (target vs. distractor) on the RIFT signal was assessed using a nonparametric dependent sample $t$ test, with the significance probability calculated by means of the Monte Carlo method. This approach involves the estimation of the null distribution by permuting the condition labels, for instance, guided search vs. unguided search 1000-10,000 times (number of permutations reported for each test). As described above, this approach was used for the identification of the RIFT sensors of interest and to contrast the RIFT response to the targets and distractors in the different conditions, for both the mean-squared coherence and GLM approach.

### Reporting summary

Further information on research design is available in the Nature Portfolio Reporting Summary linked to this article.

## Data availability

The source data to create all the figures presented in the manuscript can be found in the Supplementary Data file. Raw MEG data can be requested from the first author.

## Code availability

Custom-written analysis scripts are shared at https://github.com/katduecker/visual_search_rift

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

## Acknowledgements

This research was funded in whole, or in part, by the Wellcome Trust (227420) to O.J. Y.P. is supported by a Leverhulme Early Career Fellowship (ECF-2023-626). B.J.G. is funded by a Leverhulme Trust Early Career Fellowship (ECF-2021-628). J.W. is funded by NIH EY017001. The authors further thank Veikko Jousmaki for providing the light-to-voltage converter, Jonathan L. Winter for support with the MEG data acquisition, and Katarzyna Dudzikowska, Brandon Ingram, Alexander Murray, Davide Aloi, and Nina Salman for performing the MRI scans.

## Author contributions

S.H. proposed the research idea. K.D., K.L.S., S.H., J.W., and O.J. designed the experiment; K.D. acquired and analysed the data; Y.P. and O.J. supported analysis of MEG data; B.J.G. supervised the implementation of the GLM approach; K.D. and O.J. wrote the paper; all authors edited the paper.

## Competing interests

The authors declare no competing interests.
