## [Transparent Peer Review file · Communications Biology]

Guided visual search is associated with target boosting and distractor suppression in early visual cortex

Corresponding Author: Professor Ole Jensen

Version 0:

Reviewer comments:

Reviewer #1

(Remarks to the Author)

In this study, the authors tested whether induced oscillatory neural activity in response to invisible flicker in many-item (16 or 32-item) visual search displays (one color of items flickering at one high frequency, another color flickering at a different high frequency) tracked cued relevance of specific feature values. When the stimulus induced a measurable frequency-tagged response in the MEG signals, activity scaled with task-relevance: the target-containing stimuli were represented more strongly than distractor/non-target stimuli. There was some evidence that this might scale with task RT, suggesting some degree of behavioral relevance of these signals. The authors interpret this result as consistent with a priority map theory of visual search, because processing of target-containing stimuli is enhanced relative to distractor-containing stimuli.

While the study is interesting, there are some critical issues with its presentation at present that preclude a positive recommendation. I'm not sure I follow how the results directly inform or relate to priority map theory (as opposed to being another example of more-typical feature-based attention effects). Additionally, there are some critical methodological details missing from the current manuscript. Finally, aspects of the data presentation seem abbreviated, and additional evidence for the robustness of the frequency-tagged oscillatory signal above background/broadband frequencies would improve the manuscript. I elaborate on these issues below.

1. Reporting: The manuscript, at present, is missing a substantial number of methodological details that may be important for understanding the experiment and analysis of results. To enumerate several:
 - a. The size of the stimuli (size of each stimulus, width of letters, spatial extent of screen stimulated) is not described
 - b. The total number of blocks/trials for each participant is not clearly described
 - c. Eyetracking analysis procedures are minimally described, and do not mention what types of saccades are observed in the data (see below)
 - d. Aspects of methods describe analysis of alpha band signals (lines 491-494), which don't appear to be presented in the present manuscript
2. Relevance to priority map theory: the primary task manipulation involved pre-cueing participants about the feature value of a target stimulus, allowing them to perform 'guided search' and completely ignore distracting stimuli in another feature value. In principle, this could mean that the map of e.g. yellow stimulus locations is enhanced, while the map of blue stimulus locations is suppressed. However, it could also mean that the yellow stimuli are attended, while the blue stimuli are unattended, in the more traditional feature-based attention sense. Feature-based attention is almost surely a critical mechanism in supporting stimulus representations across priority maps, but nothing about this result speaks to priority maps themselves (unless I'm missing something). There's no ability to ascribe spatial specificity of the signals (e.g., location of a target on the screen). It's the authors' paper, and they can interpret their data within whichever framework they wish – but I do think it's worth considering exactly how this result informs priority map theory, and how this conclusion is presented in the manuscript
3. RIFT data analysis/presentation: The manuscript does not include any plots of power spectra of MEG signals to illustrate robust frequency-tagged responses. It's hard to interpret the existing data, which show changes in RIFT coherence over time, without knowing how frequency-specific the induced signal is. The Spaak et al, 2024, Imaging Neuroscience paper cited includes some great visualizations (e.g., its Fig. 3), which would be useful to see in the present work. Related, the Spaak et al 2024 Imag Neurosci paper is cited to support the somewhat-unconventional (to me) analysis choices described in the methods (e.g., compute coherence with a sine wave with white noise added, lines 381-382; generate a square wave based on sign of signal and low-pass filter to compute phase, lines 419-427, etc), but I can't find mention of these

approaches in that paper. If there's not precedent for the unconventional analysis choices in the literature, they need to be better justified either theoretically or via simulations to illustrate they are sensitive to relevant aspects of the signal. The Spaak paper appears very thorough and relatively straightforward; perhaps those metrics could be used directly instead?

4. Saccades: I appreciate that the authors have made some efforts to address the potential impact of saccades on their results, but I think it's important to present a clearer picture of what the gaze behavior looked like in this task. It seems that there was a substantial proportion of trials in which participants did not maintain fixation, as instructed. Where were they looking? How big were the saccades? How many were made on a trial, on average (and what's the distribution)? It's fair enough to include some analyses suggesting that any existing saccades don't fully account for the results, but I still think it's helpful for readers to know what the pattern of eye movements looks like in the dataset. Related, I'm curious to know if the results hold when only no-saccade trials are included in the analysis (if there are sufficiently many).

Minor:

1. Datapoints are presented with different colors in Supp Fig. 3, with no indication as to what each color means. This should be clarified.
2. The authors cite a recent paper by Thayer & Sprague (2023) as evidence for attentional modulation of BOLD, but this study, I believe, does not manipulate attention directly

Reviewer #2

(Remarks to the Author)

Reviewer #3

(Remarks to the Author)

In their manuscript entitled „Guided Visual Search is associated with a feature-based priority map in early visual cortex“, the authors present an MEG study that uses a novel technique, Rapid Invisible Frequency Tagging (RIFT), to investigate the contribution of attentional enhancement and suppression to guided search specifically and efficient attention deployment more generally. This technique also allows a localization of the effect. The authors found that both enhancement and suppression in V1 for set size 32, but not for set size 16. The results are in line with previous research showing that efficient attention deployment relies on both target enhancement and distractor suppression.

My general impression of the manuscript is very positive. The literature review is comprehensive, the methods seem sound, the results interesting and the conclusions reasonable. This is an important contribution to the field. Note: I am not an MEG expert and have never heard about the RIFT method, so I don't feel comfortable judging the method part. Unfortunately, the tight time schedule for this journal makes it difficult to really dive into the topic.

(1) I think the timing of the difference between guided / non-guided attention could be discussed in more detail. It is interesting that suppression starts earlier than the enhancement. This is related to the debate in EEG literature under which circumstances the Pd component (or Ppc) comes before the N2pc component, which seems to be context-dependent. Suppression can be reactive, i.e., a response to preceding attentional capture (Sawaki & Luck, 2013). Suppression can also be proactive, i.e. a measure to prevent attentional capture (Gaspelin & Luck, 2018), even when the exact color is not known (Feldmann-Wustefeld, Busch, & Schubö, 2019). The amount of active suppression can also predict performance (Feldmann-Wustefeld & Vogel, 2021) – it would be interesting if this could also be shown with the RIFT technique. For example, it could be that the difference between guided search and non-guided search varies as a function of behavioral performance. Alternatively, the latency may also be dependent on performance. This could be analyzed with a correlation or median split approach.

(2) The authors ran some control analyses to exclude the possibility that the observed modulation of the RIFT responses results from an eye movement bias towards the target. The main RIFT analysis is based on occipital areas, which makes sense. Would it not be a insightful check to analyze the MEG in frontal areas to see if those sensors would produce a similar pattern than that reported in Fig. 2c? If they don't, this would strengthen the authors' argument that the result is only due to differences in sensory areas and not confounded by eye-related artifacts.

(3) The authors argue that no difference was found for set size 16 compared to 32 due to too low signal-to-noise ratio. Alternatively, is it possible that we are looking at some kind of ceiling effect (enhancement and suppression so easy in the set size 16 condition that the neural signature of both processes are not visible)? In the set size 16 condition, targets and distractors are further apart than in the set size 32 condition. It was found that the N2pc (indicative of attentional enhancement) is larger when distractors are near targets (Luck et al., 1997), and analogously, the Pd (indicative of attentional suppression) is larger when when distractors are near targets (Taylor & Feldmann-Wustefeld, 2024). Reducing the number of target while keeping the number of pixels identical (for signal to noise ratio) and keeping the distance (for proximity effects on attentional processes) identical might be an approach to get less ambiguous results.

References mentioned in review:

Feldmann-Wüstefeld, T., Busch, N. A., & Schubö, A. (2019). Failed suppression of salient stimuli precedes behavioral errors. *Journal of cognitive neuroscience*, 32(2), 367-377.

Feldmann-Wüstefeld, T., & Vogel, E. K. (2019). Neural evidence for the contribution of active suppression during working memory filtering. *Cerebral Cortex*, 29(2), 529-543.

Gaspelin, N., & Luck, S. J. (2018). Combined electrophysiological and behavioral evidence for the suppression of salient distractors. *Journal of cognitive neuroscience*, 30(9), 1265-1280.

Luck, S. J., Girelli, M., McDermott, M. T., & Ford, M. A. (1997). Bridging the gap between monkey neurophysiology and human perception: An ambiguity resolution theory of visual selective attention. *Cognitive Psychology*, 33, 64–87.

Sawaki, R., & Luck, S. J. (2013). Active suppression after involuntary capture of attention. *Psychonomic bulletin & review*, 20, 296-301.

Taylor, E. D., & Feldmann-Wüstefeld, T. (2024). Reward-modulated attention deployment is driven by suppression, not attentional capture. *NeuroImage*, 299, 120831.

Zhao, G., Chen, J., Duan, Y., Li, S., Wang, Q., & Li, D. (2024). The proactive and reactive mechanisms of learned spatial suppression. *Cerebral Cortex*, 34(8), bhae333.

Version 1:

Reviewer comments:

Reviewer #1

(Remarks to the Author)

I appreciate the authors' careful attention to my previous comments and think the manuscript is greatly improved by the changes. I have no further reservations, and support its publication.

Reviewer #2

(Remarks to the Author)

The authors have addressed all the issues I raised in a satisfying way. I have no further suggestions.

Reviewer #3

(Remarks to the Author)

Reviewer #1

In this study, the authors tested whether induced oscillatory neural activity in response to invisible flicker in many-item (16 or 32-item) visual search displays (one color of items flickering at one high frequency, another color flickering at a different high frequency) tracked cued relevance of specific feature values. When the stimulus induced a measurable frequency-tagged response in the MEG signals, activity scaled with task-relevance: the target-containing stimuli were represented more strongly than distractor/non-target stimuli. There was some evidence that this might scale with task RT, suggesting some degree of behavioral relevance of these signals. The authors interpret this result as consistent with a priority map theory of visual search, because processing of target-containing stimuli is enhanced relative to distractor-containing stimuli.

While the study is interesting, there are some critical issues with its presentation at present that preclude a positive recommendation. I'm not sure I follow how the results directly inform or relate to priority map theory (as opposed to being another example of more-typical feature-based attention effects). Additionally, there are some critical methodological details missing from the current manuscript. Finally, aspects of the data presentation seem abbreviated, and additional evidence for the robustness of the frequency-tagged oscillatory signal above background/broadband frequencies would improve the manuscript. I elaborate on these issues below.

We thank the reviewer for their time and thorough evaluation of our manuscript. The reviewer has pointed to several important details that we agree needed clarification and improvement. Importantly, the reviewer's comments prompted us to a filtering issue in our RIFT analysis, which we have now updated. We have also changed the single-trial quantification of the RIFT analysis to a more conventional approach which has crystallized our results. We have also added the requested details on the eye tracking analysis. Lastly, we have modified our claims on how priority-based mapping is implemented to suggest that this mechanism is facilitated by V1 in reciprocal combination with higher order cognitive areas. All changes to the manuscript based on the reviewer's comments are marked in dark green.

Please see below for details.

1. Reporting: The manuscript, at present, is missing a substantial number of methodological details that may be important for understanding the experiment and analysis of results. To enumerate several:

a. The size of the stimuli (size of each stimulus, width of letters, spatial extent of screen stimulated) is not described

These details are now added to line 388 following in the manuscript.

b. The total number of blocks/trials for each participant is not clearly described

These details are now added to line 371 and following in the manuscript.

c. Eyetracking analysis procedures are minimally described, and do not mention what types of saccades are observed in the data (see below)

We have added the requested details to line 406 and following in the manuscript.

d. Aspects of methods describe analysis of alpha band signals (lines 491-494), which don't appear to be presented in the present manuscript

Thank you, we have deleted the paragraph.

2. Relevance to priority map theory: the primary task manipulation involved pre-cueing participants about the feature value of a target stimulus, allowing them to perform 'guided search' and completely ignore distracting stimuli in another feature value. In principle, this could mean that the map of e.g. yellow stimulus locations is enhanced, while the map of blue stimulus locations is suppressed. However, it could also mean that the yellow stimuli are attended, while the blue stimuli are unattended, in the more traditional feature-based attention sense. Feature-based attention is almost surely a critical mechanism in supporting stimulus representations across priority maps, but nothing about this result speaks to priority maps themselves (unless I'm missing something). There's no ability to ascribe spatial specificity of the signals (e.g., location of a target on the screen). It's the authors' paper, and they can interpret their data within whichever framework they wish – but I do think it's worth considering exactly how this result informs priority map theory, and how this conclusion is presented in the manuscript

We thank the reviewer for raising these important points. We decided to interpret our results in the context of the priority map framework, as several studies have suggested that the RIFT signal is strongest in early visual cortex (Zhigalov and Jensen 2019, Duecker et al., 2021; Schneider et al., 2023, Minarik et al., 2024), suggesting that the RIFT response is retinotopically organized. However, we agree with the reviewer that our data do not allow any conclusions about whether the priority map mechanism solely lives in V1, whether it lives in a higher-order area and modulates excitability in V1, or whether the mechanism emerges as the product of several mechanisms in different visual brain regions. We have now modified our description and instead argue that early visual cortex forms part of a network involving higher order cognitive brain regions to facilitate the priority-map-based mechanism.

We have rewritten the Discussion to make our points clearer. As reviewer 2 had similar concerns, we have marked the respective sections in blue, line 297 and following. The reviewer will also acknowledge that we have changed our title to "Guided visual search is associated with target boosting and distractor suppression in early visual cortex".

3. RIFT data analysis/presentation: The manuscript does not include any plots of power spectra of MEG signals to illustrate robust frequency-tagged responses. It's hard to interpret the existing data, which show changes in RIFT coherence over time, without knowing how frequency-specific the induced signal is. The Spaak et al, 2024,

Imaging Neuroscience paper cited includes some great visualizations (e.g., its Fig. 3), which would be useful to see in the present work. Related, the Spaak et al 2024 Imag Neurosci paper is cited to support the somewhat-unconventional (to me) analysis choices described in the methods (e.g., compute coherence with a sine wave with white noise added, lines 381-382; generate a square wave based on sign of signal and low-pass filter to compute phase, lines 419-427, etc), but I can't find mention of these approaches in that paper. If there's not precedent for the unconventional analysis choices in the literature, they need to be better justified either theoretically or via simulations to illustrate they are sensitive to relevant aspects of the signal. The Spaak paper appears very thorough and relatively straightforward; perhaps those metrics could be used directly instead?

We thank the reviewer for the thorough evaluation of our methods, which prompted us to rework our signal processing approach (see Rev Fig. 1). We initially chose a ± 5 Hz Butterworth bandpass filter, as it resulted in the least amount of ringing (see impulse response functions of 3 different Butterworth filters in Rev Fig. 1a). However, the reviewer's comment on the frequency specificity prompted us to a filtering artefact at ~ 63.5 Hz when using a ± 5 Hz Butterworth filter (Rev. Fig. 1B, C). We therefore opted to change our filter from a ± 5 Hz Butterworth, to a ± 3.5 Hz window-sinc Finite Impulse Response filter (FIRWS). As depicted in Rev Fig. 1D, this filter resulted in even less ringing than the ± 5 Hz Butterworth filter applied before. We tested the performance of the filter against the original ± 5 Hz Butterworth on a 60 Hz sine wave with added noise (Rev Fig. 1D). The FIRWS filter significantly outperformed the Butterworth filter at extracting the original signal, as indicated by a higher correlation between the original sine wave and the filtered signal. As shown in Figure 2b and c of the main manuscript, this filter further resulted in a good frequency resolution, with clear peaks at the stimulation frequencies. The main results presented in Fig. 2 still hold: For set size 32, there is a significantly stronger response to the Target colour in the guided compared to unguided search condition, and a significantly reduced response to the Distractor colour. Interestingly, we now also find a significant suppression of the RIFT response to the Distractor colour for set size 16. We discuss the implications of this finding in the revised manuscript.

Rev. Fig. 1A | The impulse response functions for the Butterworth filter at 3 different frequencies. In the original version of the manuscript, a ± 5 Hz filter was chosen as it resulted in the least amount of ringing in the impulse response function. B | The RIFT responses quantified by the magnitude-squared coherence obtained with a ± 5 Hz Butterworth filter and Hilbert transform. There appears to be a spurious peak between the two target frequencies. C | The Time-Frequency Representation of coherence indicates a filter artefact at 63.5 Hz. D | Comparison of the newly selected FIRWS filter at ± 3.5 Hz and the original Butterworth filter. The FIRWS filter causes less ringing in the impulse response function. E | Filter performance on a 60 Hz sine wave, as a function of added noise level. The FIRWS filter reproduces the original 60 Hz sine wave from a noisy signal reliably better than the Butterworth filter, as indicated by a higher correlation between the filtered signal and original sine wave ($T(38) = 2.25$, $p = 0.03$).

The reviewer further commented on the noise that was added to the perfect sine wave that replaced the photodiode signal. We have now changed our analysis such that the noise was only added when we calculated the coherence between the MEG sensors and RIFT response at frequencies other than 60 and 67 Hz, shown in Figure 2b and c. This was necessary, as the computation of the mean-squared coherence with Hilbert transform involves a division of the power of the respective signals (see equation 3 in the manuscript and line 492 and following). For the perfect sinusoid used to represent the photodiode, the power is 0 outside 60 and 67 Hz. Adding low-amplitude noise with a magnitude of 0.05 prevented division by 0 for frequencies different from 60 and 67 Hz, allowing us to illustrate the time-frequency representation of power and power spectra.

Thanks to the reviewer's thorough comments on our methods, we have updated the estimate of the coherence at the single trial level. We now use an FFT-based approach with overlapping sliding windows. The method and results are presented in lines 185 and line 493 and following, as well as Figure 3 of the manuscript. As the reviewer will appreciate, the results are in line with the magnitude-squared coherence approach presented in Figure 2. We could not fully follow the approach of Spaak et al 2024, as it required tagging at different phases, and spatially separable stimuli.

4. Saccades: I appreciate that the authors have made some efforts to address the potential impact of saccades on their results, but I think it's important to present a

clearer picture of what the gaze behavior looked like in this task. It seems that there was a substantial proportion of trials in which participants did not maintain fixation, as instructed. Where were they looking? How big were the saccades? How many were made on a trial, on average (and what's the distribution)? It's fair enough to include some analyses suggesting that any existing saccades don't fully account for the results, but I still think it's helpful for readers to know what the pattern of eye movements looks like in the dataset. Related, I'm curious to know if the results hold when only no-saccade trials are included in the analysis (if there are sufficiently many).

We thank the reviewer for raising these important points and agree that we should have provided more detail for the control analyses on the eye tracking data. We have now dedicated a whole section to these control analyses, please see from line 241. Moreover, we added a heatmap to Supplementary Figure 3, showing that the identified eye movements reflected horizontal microsaccades, largely within 1 degree visual angle of the fixation dot. Unfortunately, due to the long duration of the baseline and search period, it was not possible to discard trials with eye movements. As we outline in the manuscript, fixational eye movements have been argued to serve to avoid visual fading caused by neural adaptation to a stabilized retinal image (Liang et al., 2005; Martinez-Conde et al., 2004). We hope that the added details on the analysis of the eye tracking and the heatmap will satisfy the reviewer's concerns.

Minor:

1. Datapoints are presented with different colors in Supp Fig. 3, with no indication as to what each color means. This should be clarified.

Thank you, we have added a legend to the figure.

2. The authors cite a recent paper by Thayer & Sprague (2023) as evidence for attentional modulation of BOLD, but this study, I believe, does not manipulate attention directly

Thank you, we have removed this reference.

Reviewer #2 (Remarks to the Author):

This study investigates neural processes underlying feature guidance in visual search with MEG. Subjects search for a shape defined target among distractors drawn in two different colors. The color of the target is either cued, such that search can be restricted to a subset of items (color guidance), or the cue is non-informative requiring a search among all items of the array (no color guidance). The search is performed with two set-sizes of 16 or 32 items. To assess the guidance-related neural activity, a stimulation protocol - referred to as rapid invisible frequency tagging (RIFT) – is used, which allows for separating the cortical response to the two colors. It is observed that for both set-sizes a clear behavioral guidance effect by color is seen, with the RIFT response in early visual cortex being enhanced for the target color, but attenuated for the non-target color. This guidance-associated RIFT modulation, however, is not observed for the set-size of 16 items. The authors conclude that the early visual cortex (V1) contributes to a ‘priority- map based mechanism’ underlying color guidance in visual search.

This study investigates the neural mechanisms behind feature guidance in visual search by employing a tried and tested experimental approach. The behavioral results are convincing and confirm color guidance effects shown in numerous previous studies. The MEG-based RIFT results are less clear, as they are not consistent across the set-size conditions, and the authors’ explanation of the discrepancy in terms of low signal-to-noise is not compelling. These issues are detailed below.

We thank the reviewer for the detailed evaluation of our manuscript, that has resulted in several improvements to the paper. We have marked all changes that address the reviewers concerns in magenta.

(1) The set-size 16 shows no RIFT modulation as a function of color guidance which is a major issue. When comparing set-size 16 and 32 the RT decrement for the guided versus the unguided search is roughly the same, and the statistical validation confirms the presence of a guidance effect for both set-sizes. If the RIFT data do not show an effect in the set-size 16 condition, why should one assume that the effect seen for set-size 32 reflects color guidance? As an account, the authors argue that the signal-to-noise is too low at this set-size because a fewer number of pixels flicker. This does not convince as an account. Compare the data shown in Figure 2b and c. For set-size 16, the modulation difference between the unspecific peak at ~80ms and the subsequent minimum around 200ms is roughly ~60% of that of set-size 32. For set-size 32, the size of the RIFT modulation by guidance (target versus distractor after 300ms) is roughly 50% of the unspecific modulation peak. Hence, a 50% modulation of the 80ms peak of the set-size 16 condition should be very apparent! Furthermore, the target enhancement is the largest modulation in the set-size 32 condition. If a difference in signal-to-noise accounts for the discrepancy between set sizes, why is it that we see a small, apparently non-significant, distractor attenuation for set-size 16 instead of a small target enhancement? Finally, the signal-to noise account is problematical in another respect. It implies that RIFT is not sensitive to all set-sizes. But then, it may not be a reliable measure of feature guidance to begin with. The authors may want to think about other explanations. For example, the item density in set-size 32 is higher than in set-size 16, requiring a

higher spatial resolution of selectivity, which emphasizes feature-guidance to operate in early visual cortex. For set-size 16, the resolution in mid-level areas may suffice to select the target, such that guidance-related modulations are best applied here.

We understand the reviewer's concerns and agree that the points we had made regarding the signal-to-noise ratio should be clarified. Based on the comments by reviewer 1, we have made several changes to our signal processing that improved the extraction of the RIFT response. We now find a significant reduction of the response to the distractors for guided compared to unguided search (see Figure 2d). Based on reviewer 3's comments, we now discuss further explanations on why the set size 16 RIFT results are less robust than the set size 32 results, that go beyond the signal-to-noise ratio account. Please see line 351 and following (marked in purple).

(2) Discussion p9 259-262: Here the authors conclude that they have provided evidence for a retinotopically organized priority map in early visual cortex. I am afraid that this conclusion is beyond what the data show. There is nothing in the data that would speak to the question whether the cortical response assessed with RIFT is retinotopically organized or not. What's is presumably seen in the set-size 32 condition is that the target color is globally enhanced (distractor color attenuated) at all locations where it happens to be present. As the target and distractor color are randomly distributed in the same area of the search array, we don't know whether the color bias is retinotopically selective. For this to clarify one would have to compare color responses at different locations, say in the left versus right visual field. A more principal issue. The authors annotate that RIFT does not propagate beyond V1/2. If this is the case, it cannot speak to the question where the priority map is located in the first place, as it does not 'see' activity from a priority map eventually located elsewhere (say in V4, LIP, frontal areas).

We agree with the reviewer's concern that we could have nuanced these results better in the manuscript. Our main argument for a priority-map based mechanism is that the modulation of the neuronal excitability affects primary visual cortex, which is retinotopically organized and thus has a high spatial resolution. We have updated several paragraphs in the discussion to address the reviewer's comment. This section emphasizes that our results speak to a modulation of the excitability of retinotopically organized early visual neurons responding to different locations in parallel based on their colour-preference. This finding is not trivial as it suggests that early visual cortex in conjunction with higher order cognitive brain regions are responsible for guiding visual search. We agree that this finding does not provide evidence on the location of the priority map, yet it suggests that early visual cortex partakes in its implementation. The respective paragraphs are marked in blue, line 291 and following in the manuscript as they also address concerns raised by reviewer 1. As the reviewer will acknowledge, we have also changed the title of the manuscript.

Minor

Intro p.2 55-58: It is a bit odd to point out that recent studies have shown that early visual cortex is modulated by attention. There is a broad and comparably old

literature showing that visual attention modulates activity in V1 with EEG, MEG, and fMRI (Martinez et al. 1999 Nat.Neurosci. Noesselt et al. 2002, Neuron, Pogoshyan & Ioannides, 2008, Neuron, Kelly et al. 2008, Cereb. Cortex, DiRusso et al. 2003, Cereb.Cortex), fMRI (Tootell et al. 1998, Neuron, Brefczynski & DeYoe, 1999, Nat.Neurosci.).

The reviewer is right that these studies should be mentioned in our introduction. We have adjusted the paragraph to clarify the gap in the literature and outline the aim of our study. Please see line 51 and following in the introduction.

The topography of the GLM analysis suggests significant modulation differences (regressor contrasts) outside V1 (Figure 3b and c) as compared to what we see in Figure 2a. Does this indicate that under this analysis activity arises from extrastriate areas?

Based on another comment by reviewer 1 we have updated our estimate of the single-trial coherence used for the GLM approach (please see line 171 and following in the manuscript). The effect is now more clearly localized over occipital regions. As the cluster-based permutation t-test does not test the spatial extension of the cluster, it does not serve to make any claims on whether extrastriate areas might contribute to the effect.

Figure 3: No mention of panel (c) in the figure legend.
Thank you for catching this, we have updated the figure.

Reviewer #3 (Remarks to the Author):

In their manuscript entitled „Guided Visual Search is associated with a feature-based priority map in early visual cortex“, the authors present an MEG study that uses a novel technique, Rapid Invisible Frequency Tagging (RIFT), to investigate the contribution of attentional enhancement and suppression to guided search specifically and efficient attention deployment more generally. This technique also allows a localization of the effect. The authors found that both enhancement and suppression in V1 for set size 32, but not for set size 16. The results are in line with previous research showing that efficient attention deployment relies on both target enhancement and distractor suppression.

My general impression of the manuscript is very positive. The literature review is comprehensive, the methods seem sound, the results interesting and the conclusions reasonable. This is an important contribution to the field. Note: I am not an MEG expert and have never heard about the RIFT method, so I don't feel comfortable judging the method part. Unfortunately, the tight time schedule for this journal makes it difficult to really dive into the topic.

We thank the reviewer for their time evaluating our work and the positive feedback. All changes made to the text based on the reviewer's remarks are marked in orange.

(1) I think the timing of the difference between guided / non-guided attention could be discussed in more detail. It is interesting that suppression starts earlier than the enhancement. This is related to the debate in EEG literature under which circumstances the Pd component (or Ppc) comes before the N2pc component, which seems to be context-dependent. Suppression can be reactive, i.e., a response to preceding attentional capture (Sawaki & Luck, 2013). Suppression can also be proactive, i.e. a measure to prevent attentional capture (Gaspelin & Luck, 2018), even when the exact color is not known (Feldmann-Wustefeld, Busch, & Schubo, 2019). The amount of active suppression can also predict performance (Feldmann-Wustefeld & Vogel, 2021) – it would be interesting if this could also be shown with the RIFT technique. For example, it could be that the difference between guided search and non-guided search varies as a function of behavioral performance.

Alternatively, the latency may also be dependent on performance. This could be analyzed with a correlation or median split approach.

Based on the evaluation by reviewer 1, we have updated the coherence approach to quantify the RIFT response, which no longer implies that the distractor suppression precedes target boosting. As the cluster-based permutation test does not explicitly test the temporal extent or onset of the cluster, we are hesitant to make strong claims about the differences in the onset of the target boosting and distractor suppression. Importantly, we did not aim to study anticipatory distractor suppression and have therefore not optimized our experimental design to do so (we have now outlined this in the discussion, line 332 and following). Interestingly, however, with the updated methods, we do find evidence for distractor suppression but not target boosting in the set size 16 guided search condition (see Figure 2d). We also performed a median split of the RIFT signal based on reaction time and found no

difference for fast vs slow trials (see line 236 following, and Supplementary Figure 2).

(2) The authors ran some control analyses to exclude the possibility that the observed modulation of the RIFT responses results from an eye movement bias towards the target. The main RIFT analysis is based on occipital areas, which makes sense. Would it not be a insightful check to analyze the MEG in frontal areas to see if those sensors would produce a similar pattern than that reported in Fig. 2c? If they don't, this would strengthen the authors' argument that the result is only due to differences in sensory areas and not confounded by eye-related artifacts.

This is a good point that we have now further clarified in Figure 3 and the associated text. The cluster-based test for the GLM approach included all sensors and revealed a modulation of the RIFT response in the occipital sensors, but not the frontal sensors. This is likely due to the RIFT signal not propagating meaningfully beyond early visual cortex (as we outline in the manuscript line 290). This paragraph addresses concerns raised by all reviewers and is therefore marked in blue.

(3) The authors argue that no difference was found for set size 16 compared to 32 due to too low signal-to-noise ratio. Alternatively, is it possible that we are looking at some kind of ceiling effect (enhancement and suppression so easy in the set size 16 condition that the neural signature of both processes are not visible)? In the set size 16 condition, targets and distractors are further apart than in the set size 32 condition. It was found that the N2pc (indicative of attentional enhancement) is larger when distractors are near targets (Luck et al., 1997), and analogously, the Pd (indicative of attentional suppression) is larger when when distractors are near targets (Taylor & Feldmann-Wustefeld, 2024). Reducing the number of target while keeping the number of pixels identical (for signal to noise ratio) and keeping the distance (for proximity effects on attentional processes) identical might be an approach to get less ambiguous results.

Thanks to a comment by reviewer 1, we have now updated our coherence analysis and show that there is indeed evidence for significant distractor suppression in the set size 16 condition. However, the RIFT results in the set size 16 condition are still not as robust as the ones presented for set size 32. The reviewer is correct that insufficient signal-to-noise ratio is only one potential explanation for the absence of the effect. We thank the reviewer for pointing us to the relevant findings in the EEG/ERP literature. We now dedicate a new paragraph in the Limitations & Outlook sections to additional explanations, including the proximity of target and distractor stimuli. This paragraph also addresses concerns raised by reviewer 2 and is therefore marked in purple in line 351 and following in the manuscript.